# Audio Denoising Coprocessor Based on RISC-V Custom Instruction Set Extension

**Jun Yuan [1], Qiang Zhao [1,\*], Wei Wang [1], Xiangsheng Meng [1], Jun Li [1] and Qin Li [2]**

[1]  School of Optoelectronic Engineering, Chongqing University of Posts and Telecommunications, Chongqing 400065, China; yuanjun@cqupt.edu.cn (J.Y.); wangwei@cqupt.edu.cn (W.W.); s200431041@stu.cqupt.edu.cn (X.M.); s190431050@stu.cqupt.edu.cn (J.L.)
[2]  Chongqing Marketing Department, Southwest Oil & Gas Field Company, Chongqing 401120, China; liqin1@petrochina.com.cn
\*  Correspondence: s200431068@stu.cqupt.edu.cn

**Abstract:** As a typical active noise control algorithm, Filtered-x Least Mean Square (FxLMS) is widely used in the field of audio denoising. In this study, an audio denoising coprocessor based on Retrenched Injunction System Computer-V (RISC-V), a custom instruction set extension was designed and a software and hardware co-design was adopted; based on the traditional pure hardware implementation, the accelerator optimization design was carried out, and the accelerator was connected to the RISC-V core in the form of coprocessor. Meanwhile, the corresponding custom instructions were designed, the compiling environment was established, and the library function of coprocessor acceleration instructions was established by embedded inline assembly. Finally, the active noise control (ANC) system was built and tested based on Hbird E203-Core, and the test data were collected through an audio analyzer. The results showed that the audio denoising algorithm can be realized by combining a heterogeneous System on Chip (SoC) with a hardware accelerator, and the denoising effect was approximately 8 dB. The number of instructions consumed by testing custom instructions for specific operations was reduced by approximately 60%, and the operation acceleration effect was significant.

**Keywords:** RISC-V; custom instruction; ANC; coprocessor

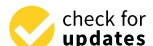



## 1. Introduction

With the rapid development of the economy, noise problems such as industrial noise and automobile noise have become increasingly prominent. The traditional passive noise control technology [1] is effective for medium and high-frequency noise through passive control methods such as sound absorption and sound insulation. However, active noise control [2], namely the method of reducing a noise signal through the principle of destructive interference of sound waves, is more effective for low-frequency narrow-band noise. An adaptive denoising system is generated; that is, while generating anti-noise signals, active repair of anti-noise signals is carried out according to the change in noise to complete active noise control.

The FxLMS algorithm is widely used in active noise control systems due to its simple circuit structure, simple implementation, and small computation. Traditional ANC implementations usually involve Digital Signal Processing (DSP) due to their certain hardware computing units and capabilities to modify software, making ANC algorithms highly flexible without impeding computing speed, but DSP chips are costly and of high power consumption [3]. Therefore, some designs suggest using a universal Micro Control Unit (MCU) as a substitute for DSPs [4]. However, universal MCUs are not only subject to the serial instruction streams but also do not have a certain hardware computing units, making ANC applications difficult due to their computing inefficiency. Reference [4] proposes an implementation of the FxLMS algorithm using a STM32F407 microprocessor of Cortex-M4

in order to improve the computing efficiency, and proposes a fixed step size method to reduce the computation and solve the problem of floating-point operation.

Field programmable gate arrays (FPGAs), with their highly parallel computing capabilities, are also used for ANC implementations [5]. FPGAs are designed totally based on hardware logic, so they require no software instructions for control as DSPs and MCUs do, making FPGAs unconstrained from serial instruction streams; in turn, their parallel processing capabilities speed up the computing process. Reference [5] proposed a hardware implementation of the FxLMS algorithm based on FPGA, which divides the operation part of the algorithm into the filtering part and update part; in which the filtering part is FIR filter, namely the process of one-dimensional convolution, and the update part is the weight update part of LMS algorithm block, which is the process of multiply accumulate (MAC). Using FPGAs as the hardware of ANC algorithms significantly improves the computing capacity, but the flexibility of the algorithms is sharply reduced because the hardware design forces the algorithms to redesign the hardware logic for a slight change. In addition, application-specific integrated circuit (ASIC) [6] and analog circuit [7] are used to realize the ANC system, but the overall effect is not as good as the mainstream DSP or FPGA. Therefore, a new design with the advantages of both DSPs and FPGAs was developed; that is, a combination of RISC-V soft core with a dedicated hardware accelerator. An audio denoising coprocessor based on RISC-V custom instruction set extension was designed in this study. According to the hardware implementation of the traditional FxLMS algorithm, the software and hardware co-design of the FxLMS algorithm was carried out, the work of filling and moving the data to be processed were handed over to MCU for processing. Meanwhile, the convolution and MAC operations with large computation were designed as hardware accelerators, and the coprocessor was designed in the way of instruction pipeline. Finally, the hardware acceleration was completed by the coprocessor, and the heterogeneous SoC was combined with the hardware accelerator.

The new method ensures design flexibility and controllability, while the dedicated hardware accelerator maintains highly parallel computing. Furthermore, the RISC-V soft core designs the mounted hardware accelerator in the form of a co-processor, which makes full use of pipelining techniques.

## 2. RISC-V and Hbird E203 Core

The RISC-V instruction set has been widely welcomed all over the world since it was published in 2014. The RISC-V instruction set design is simplified and efficient [8]. At present, the setting of RISC-V modular instruction set makes RISC-V architecture have more choices, so that it can attempt to meet various applications through a unified architecture, which is an advantage that X86 and the ARM instruction set architecture do not possess [9]. Extensibility of instructions is a prominent feature of RISC-V architecture. Users can customize instructions according to the reserved instruction coding space so the coprocessor has better portability.

In order to realize the audio denoising coprocessor based on the RISC-V instruction set, it is necessary to select the appropriate RISC-V processor core as the carrier. Among many open-source RISC-V cores, such as Rocket_Core [10], BOOM_Core [11], RI5CY_Core [12], and others, Hbird E203 core adopts two-stage pipeline design and supports RV32I/E/A/M/C instruction subset configuration, and its supporting SoC provides a large number of Intellectual property (IP) core, including Universal Asynchronous Receiver Transmitter (UART), Inter Integrated Circuit (IIC), Serial Peripheral Interface (SPI), etc., [13]. Due to the rich SoC resources of the E203 core and its mature design tools, this study selected the E203 core as the processor core of this design. ARM Cortex-M0+ was used as the benchmark in terms of performance, and its microarchitecture is shown in Figure 1.

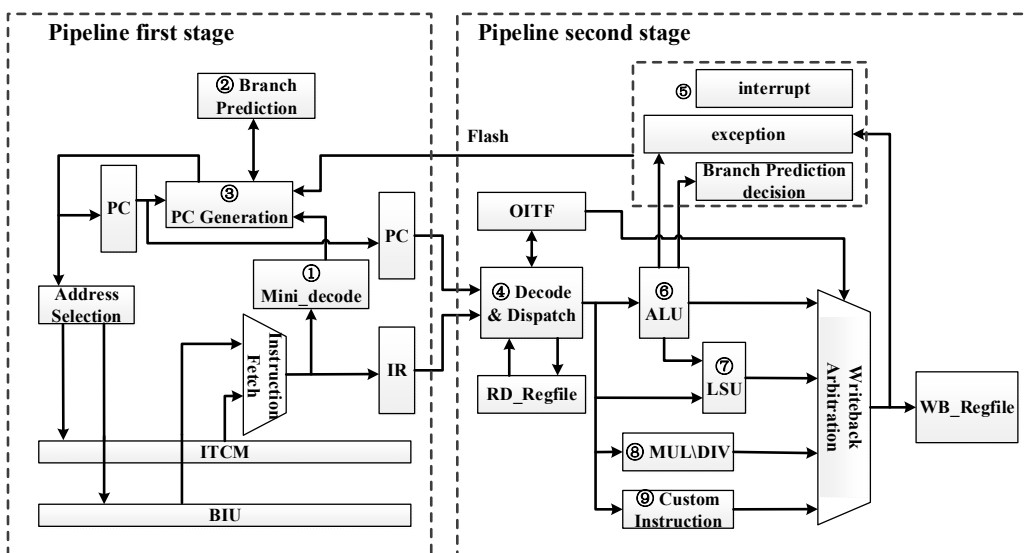

**Figure 1.** Schematic diagram of E203 microarchitecture.

E203 core adopts a two-stage pipeline structure, the first stage of which is instruction fetch (IF), and the second stage of which is instruction decode (ID), execute (EX), writeback (WB), and memory (MEM).

The first stage pipeline includes a simple ID function block, branch predictor and program counter (PC) generator. The simple ID function block (tag 1 in the figure) partially decodes the obtained instructions to obtain some instruction information, including the classification of instructions, whether they are ordinary instructions or branch jump instructions, and the types and details of branch jump instructions. For branch jump instruction, it is necessary to use a static branch predictor (tag 2 in the figure) to predict the jump and obtain the predicted jump address of the instruction. The PC generator (tag 3 in the figure) generates the PC value of the next instruction to be fetched, generates the PC according to different types such as fetching after reset, sequential fetching, branch instruction fetching and pipeline flushing fetching, and accesses instruction tightly coupled memory (ITCM) or bus interface unit (BIU) to instruction fetch through internal chip bus (ICB). The PC value and the corresponding instruction value are stored in the PC register and the IR register.

The secondary pipeline mainly includes ID and dispatch (tag 4 in the figure), arithmetic logic operation unit (tag 6 in the figure), memory access unit (tag 7 in the figure), long instruction (tag 8 in the figure), custom instruction (tag 9 in the figure), delivery, and pipeline flushing (tag 5 in the figure). ID and dispatching realize ID of instructions and dispatching related information to arithmetic logic unit (ALU), and ALU unit dispatches specific information to different execution units for execution. One-cycle instructions such as logic operation, addition and subtraction, shift, etc., are handed over to ordinary ALU unit for processing. The branch jump instruction is delivered to judge the prediction, and the prediction error needs to be flushed by the instruction pipeline. The memory access instruction is allocated to the memory loading unit for loading and accessing data. Long-term coprocessor instructions are assigned to coprocessor units for execution.

## 3. The FxLMS Algorithm and Its Hardware and Software Co-Design Conception

The schematic diagram of active noise control system architecture is shown in Figure 2, and the operation processing part is the most classical FxLMS algorithm [14–16].

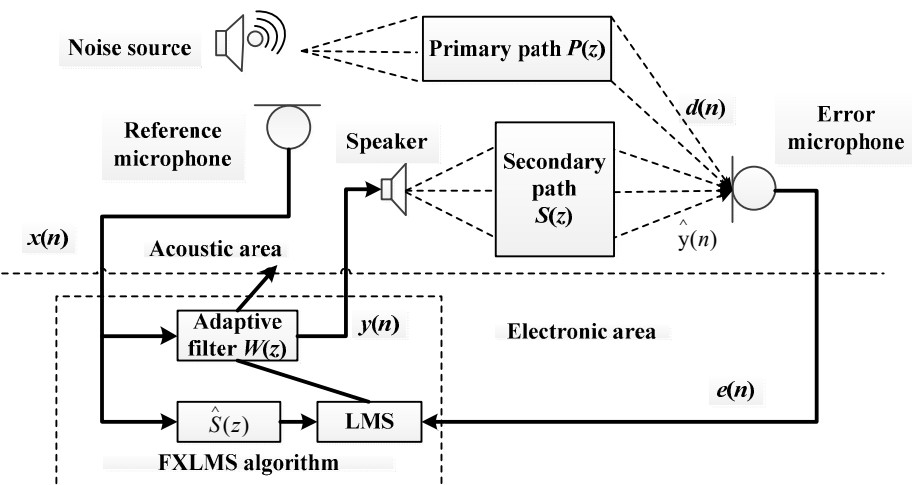

**Figure 2.** Schematic diagram of ANC system structure.

The implementation of the FxLMS algorithm has two different acoustic paths. The main signal is sampled by the audio codec module WM8731 with built-in ADC, then the speaker emits an anti-noise signal, and the error sensor measures the residual error signal. In this process, the acoustic path between the reference noise source and the error sensor is called the primary path, and the electrical to acoustic path between the speaker and the error microphone is called the secondary path. The FxLMS algorithm contains two parts, one is the least mean square algorithm, and the other is adaptive filtering.

### 3.1. LMS Algorithm Principle

The least mean square (LMS) algorithm is based on the minimum mean square error criterion and the gradient method. By improving the calculation method of the gradient value of the mean square error, the algorithm can be shown by recursive formulas such as Equations (1)–(3) [17–19]:

$$y(n) = W^H(n)X(n) \tag{1}$$

$$e(n) = d(n) - y(n) \tag{2}$$

$$W(n+1) = W(n) + 2\mu X(n)e^*(n) \tag{3}$$

where $W(n)$ represents the weight vector of the filter; $X(n)$ represents a set of vectors composed of input signals; $y(n)$ represents the output signal; $d(n)$ represents the desired signal; $e(n)$ represents the error signal; and $\mu$ represents the step size factor, where the larger $\mu$ is, the faster the convergence speed of the algorithm is, and vice versa. However, the faster the convergence speed, the worse the steady-state performance, so it is necessary to constrain the step size factor. In this design, considering the reduction in algorithm complexity and processing flexibility, the selection of step factor is based on the fixed step proposed in [4].

### 3.2. Adaptive Filtering

The adaptive filtering part is a FIR filter, and the formula is shown in Equation (4):

$$y(n) = W^T(n)X(n) \tag{4}$$

where $y(n)$ represents $X(n)$ generated by an FIR filter with a weight coefficient $W(n)$. Because every time a stage sound source $y(n)$ is generated, the weight coefficient $W(n)$ is updated by LMS operation; therefore, updated time-varying coefficients are obtained, i.e., the coefficients are automatically and continuously adapted to a given signal to obtain a desired response to complete adaptive filtering.

### 3.3. Software and Hardware Co-Design

The coprocessor part is connected with the main processor in the mode of instruction pipeline through Nuclei Instruction Co-Unit Extension (NICE) circuit interface, and the hardware acceleration function is mobilized in the mode of custom instructions in the software flow, which is shown in Figure 3.

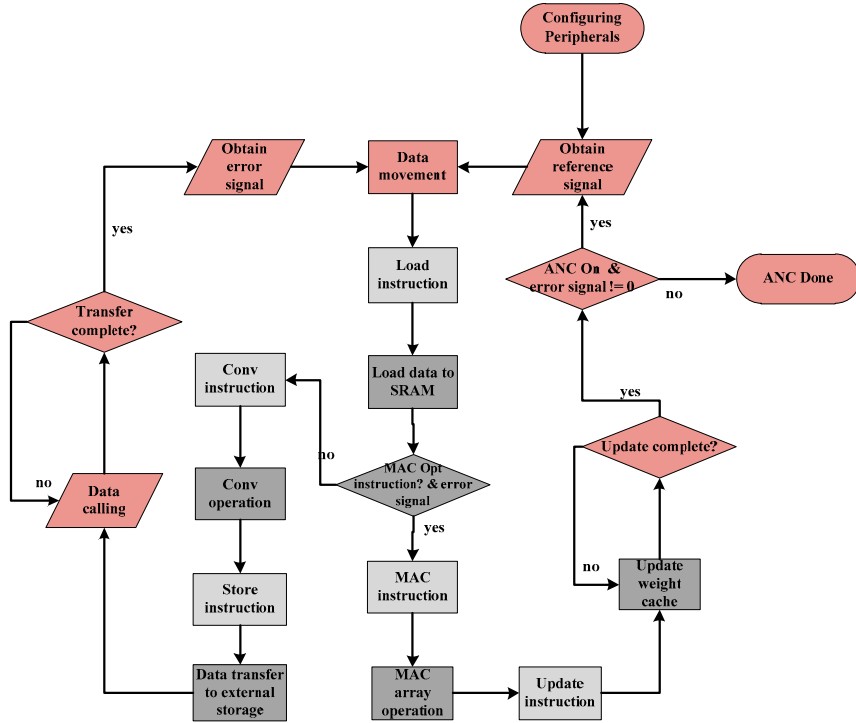

**Figure 3.** Flow chart of software and hardware co-design.

The pink parts in Figure 3 belong to the software flow, of which specific significance is the external acquisition of the ANC system and the configuration of function blocks. Then it includes storing the data at the corresponding address after collection. Subsequently, the light gray part is the related custom instructions. Used for realizing software and hardware interaction between the main processor and the coprocessor, the last dark gray part is the defined hardware acceleration part, which specifically includes the adaptive filtering part in the algorithm corresponding to convolution operation, the weight update part corresponds to the LMS algorithm in the algorithm corresponding to multiplication and accumulation array operation, and the corresponding cache unit used in data handling.

At the same time, due to the particularity of serial operation of the algorithm itself, this process has two steps. First, the black flow line is the adaptive filtering operation in the main path, and then the electrical to acoustic transformation is needed through relevant peripherals to generate secondary sound sources. The second step is the acquisition of error signals and the updating operation of weight coefficients, which have a sequence relationship. Therefore, this design process includes two paths, and only when both paths run out can the ANC system denoising be completed once.

## 4. Hardware Design Part

The audio denoising accelerator designed in this study optimizes the updating weight and filtering module in the traditional design. The parallel one-dimensional convolution structure in the form of an addition tree is used to replace the serial MAC arithmetic unit to realize the filtering part, and the parallel MAC array is used to replace the original updating weight part, and the related modules of coprocessor are added.

### 4.1. Optimization of Operation Structure Design

In the traditional hardware implementation of the FxLMS algorithm, the filter module adopts the MAC arithmetic unit, namely the multiply accumulate arithmetic unit, and realizes filtering in serial mode which reduces the arithmetic performance of the filter module, and a lot of repeated operations are needed when the filter order is long. Therefore, this design adopts the strategy of sacrificing area in exchange for performance improvement, and uses the addition tree structure, which greatly improves the parallel operation ability and realizes one-dimensional convolution operation. At the same time, this design adopts the MAC array parallel operation to update the coefficients of the weight matrix. Finally, this design adopts the idea of data multiplexing, and uses a data distributor to reduce the resource consumption of weight coefficient storage SRAM. The circuit structure is shown in Figure 4.

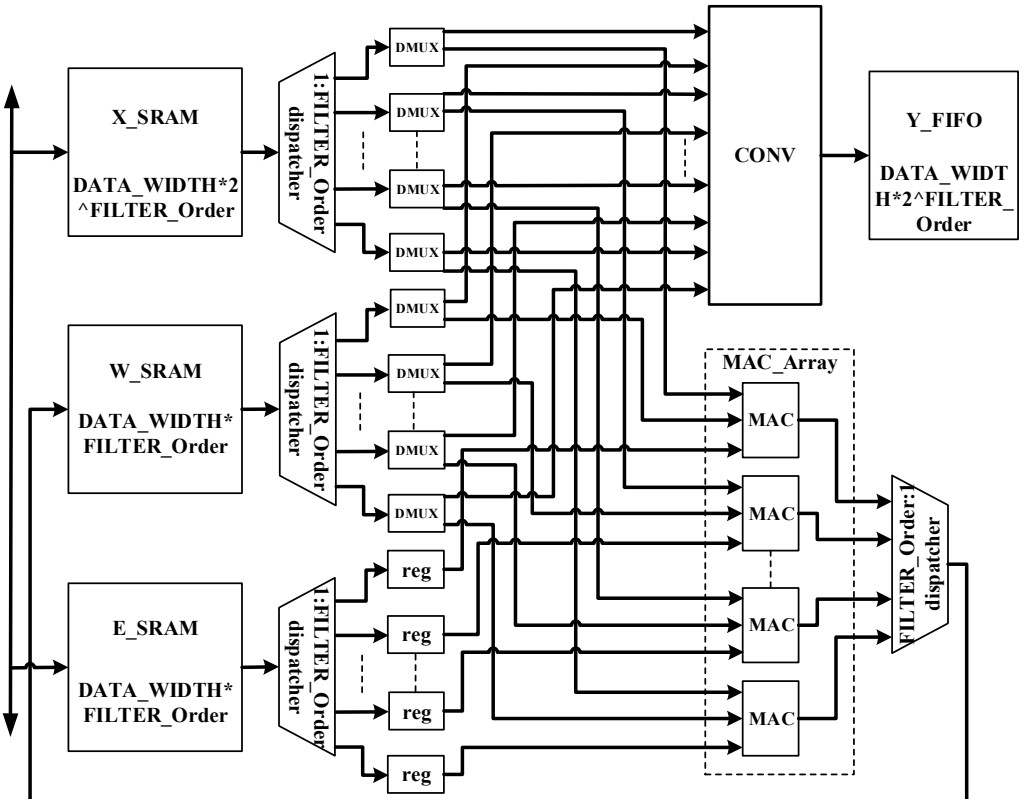

**Figure 4.** Structure diagram of hardware accelerator.

The whole acceleration circuit comprises a reference signal data buffer module (X_RAM), a weight coefficient data buffer module (W_RAM), an error signal data buffer module (E_RAM), a data distributor, a one-dimensional convolution operation block, a MAC array operation block, and a data integrator.

The most critical parts in the accelerator circuit are the one-dimensional convolution operation block and MAC array operation block. The one-dimensional convolution operation block realizes the FIR filtering part of the algorithm, and the MAC array operation block realizes the parallel weight coefficient update. Its circuit structure is shown in Figure 5.

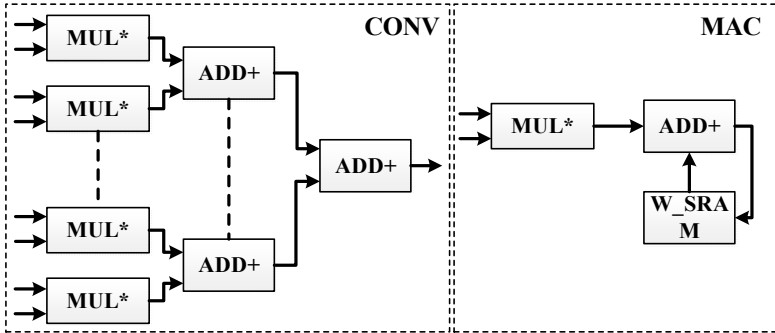

**Figure 5.** Circuit structure diagram of arithmetic unit.

Because the audio denoising algorithm needs to quickly generate secondary sound sources after collecting reference signals, and the generation of secondary sound sources must be operated through adaptive filtering; so in this design, the addition tree parallel structure is used to design the one-dimensional convolution operation for filtering, which can improve the operation speed and high parallelism so that the secondary sound sources can be produced faster. When the secondary sound source is generated, it is necessary to collect error signals to update the filter weight coefficients, so the algorithm has the characteristics of sequential processing, and the speed of weight updating plays an important role in the generation of secondary sound sources. Therefore, in this design, the MAC array is used to realize the updating operation of each weight, and the updated weight data needed by the next convolution operation can be obtained in the same period. The reasonable use of the data distributor and data integrator greatly improves the operation speed.

*4.2. Coprocessor Design*

After the operation structure design is completed, the core instruction cooperation unit should be added to expand the coprocessor design, and the decoder, data extractor and configuration enabling function block should be added to complete the hardware design of the audio denoising coprocessor. Its circuit structure is shown in Figure 6.

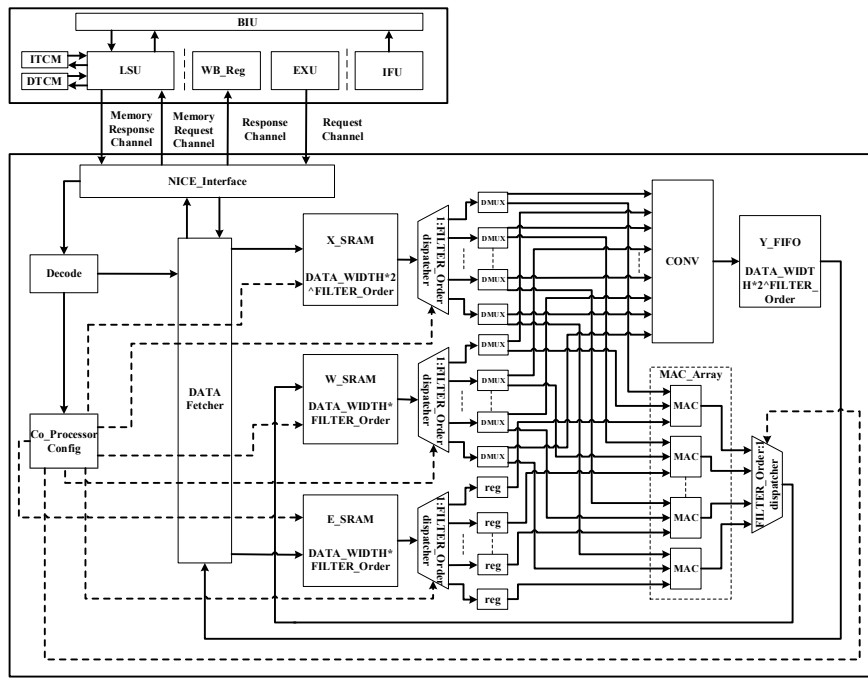

**Figure 6.** Circuit structure diagram of audio denoising coprocessor.

The NICE controller processes the time sequence related to the interface of the co-processor and transmits the instruction information and source operands obtained from the request channel to the decoder for ID. The decoder is used to decode custom instructions. This design is mainly divided into two types of instructions, one is configuration instructions, and the other is data loading and storage instructions. For the configuration instruction, the configuration information is transmitted to the configuration module, and the configuration module provides the enabling signal and control signal required by the corresponding response module to realize the configuration of each operation function part. For the data load store instruction, the memory access information is transmitted to the data extractor for processing. When the memory access information is a loading instruction, the address information and the read signal are transmitted through the memory request channel and the data are obtained from the corresponding memory module. Then the read data are transmitted to the data extractor through the memory feedback channel and distributed to the corresponding cache module by the data extractor. When the memory access information is a write memory instruction, the address information is transmitted through the memory request channel, and the write data are obtained from the data extractor and transmitted to the memory location corresponding to the address. If the instruction is the write-back result, the write-back data are transmitted to the feedback channel through the data extractor to complete the write-back of the general register.

After the software program is burned to the MCU, the main processor obtains the instructions in sequence, decodes the instructions, and judges whether the instructions are custom instructions according to their operation codes. In this design, the operation codes of custom 1–4 defined by RISC-V are used as custom instruction operation codes, and R-type instructions are used for custom instruction coding. Its format is shown in Figure 7.

**Figure 7.** The 32-bit custom instruction encoding format.

For custom instructions, it is judged whether to read the source operand according to xs1 and xs2. In this process, the main processor maintains the data correlation, and if there is a data conflict, the data channel is closed until the data correlation is released. If there are data written back, the destination register of the rd bit is also a consideration of data correlation. After that, the instruction information is transmitted to the coprocessor for processing through the NICE interface. The coprocessor decodes the instructions and distributes them to different units for execution according to the type of instructions. Finally, the coprocessor writes the instruction execution results back to the main processor through the response channel, and writes the execution results back to the rd target register or transmits the results to the corresponding storage locations through the memory request channel.

The data stream of the audio noise reduction coprocessor designed in this subject is shown in Figure 8, and the processing signals are obtained by external sensors or receivers. Data are transmitted to ICB peripheral bus through interface IP mounted on SoC. When the relevant data need to be processed, the data are acquired and written through the memory request and feedback channel. After the processing is finished, the processing result is written back to the general register of the main processor or into the corresponding memory, and the main processor sends it to the external module through the interface IP to obtain the generated signal.

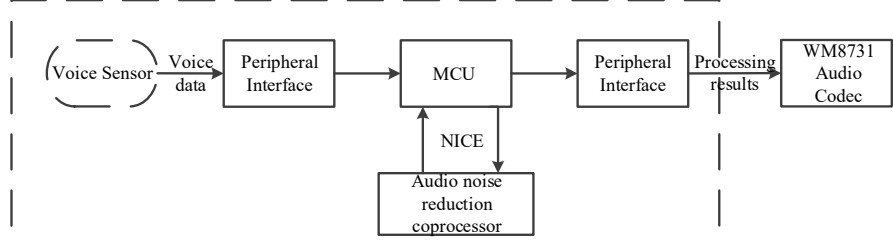

**Figure 8.** System data flow diagram.

Through memory access custom instructions, a lot of data stored in the main processor are moved to the coprocessor, which reduces the access of the coprocessor to the main processor memory and greatly reduces the power consumption. At the same time, the parallel operation units in the coprocessor ensure the operation speed. Finally, compared with the SoC plug-in accelerator, the coprocessor with instruction pipeline mode does not need frequent data access, reduces data movement, and has better real-time processing performance.

## 5. Software Design Part

*5.1. Custom Instruction Design*

The custom instructions of the audio denoising coprocessor based on the FxLMS algorithm are shown in Table 1.

**Table 1.** Custom instruction table of coprocessor.

| Instruction | Funct7 | Rd | Xd | Rs1 | Xs1 | Rs2 | Xs2 |
|---|---|---|---|---|---|---|---|
| Load.X | 1 | - | 0 | X_MemoryAddress | 1 | Length ǀ X_BaseAddress | 1 |
| Load.E | 2 | - | 0 | E_MemoryAddress | 1 | Length ǀ E_BaseAddress | 1 |
| Store.Y | 3 | - | 0 | Y_MemoryAddress | 1 | - | 0 |
| Cfg.Conv | 4 | - | 0 | Filter order | 1 | En_Conv | 1 |
| Cfg.MAC | 5 | - | 0 | Filter order | 1 | En_MAC | 1 |
| Updata.W | 6 | - | 0 | Filter order | 1 | En_Up.W | 1 |
| Rst | 7 | - | 0 | - | 0 | - | 0 |

There are seven custom instructions, namely the data load storage instruction and configuration enable instruction. The data loading instruction is responsible for loading the reference signal and the error signal from the corresponding address and storing them in the corresponding buffer of the coprocessor. The data storage instruction is responsible for transmitting the secondary sound source signal and writing it to the corresponding memory address through the memory request channel. The configuration enable instruction is responsible for configuring the filter order and enabling the relevant functional modules.

The use steps of custom instruction are shown in Figure 9: firstly, the reference signal is loaded through Load.X instruction, and the data are accessed through the memory request channel and read through the memory feedback channel, and then loaded into the X_SRAM cache. After that, the reference signal and weight coefficient are read from X_SRAM and W_SRAM by Cfg.Conv instruction and sent to the corresponding DMUX through data distributor. DMUX performs the convolution operation and generates the secondary sound source under the control of the enable signal until the convolution of reference signal ends. After that, the secondary sound source data in FIFO is written back to the corresponding address through the memory request channel through Store.Y instruction. Then Load.E instruction loads error signal data such as Load.X instruction, and Cfg.MAC instruction configures MAC operation array and updates weight coefficients. Finally, W_SRAM is configured by Updata.W instruction to write the update weight, which completes an adaptive denoising operation acceleration. In addition, the reset of the coprocessor can be performed by Rst instruction.

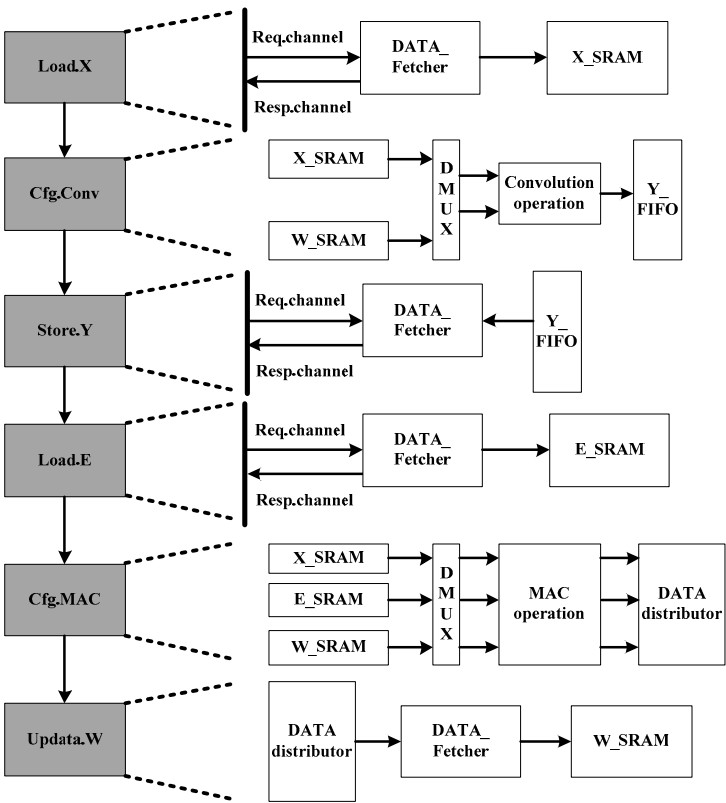

**Figure 9.** Usage flow and hardware operation of custom instruction.

### 5.2. Coprocessor Library Function Design

After completing the instruction of the custom coprocessor, we can use assembly language to transfer the work of coprocessor. However, the efficiency of assembly language development is too low, so embedded inline assembly is often used in C\C + +. Therefore, the first task is to package instructions into C language library functions by using the inline assembly syntax format and completing the library function design of coprocessor. The designed library function interface is shown in Table 2.

**Table 2.** Library functions of custom instructions and their introduction.

| Function Interface | Function |
|---|---|
| int Load_X(unsigned int X_MemoryAddress, unsigned int X_BaseAddress, unsigned int Length) | Load reference signal X into X.SRAM |
| int Load_E(unsigned int E_MemoryAddress, unsigned int E_BaseAddress, unsigned int Length) | Load error signal E into E.SRAM |
| int Store_Y(unsigned int Y_ MemoryAddress) | Store secondary source Y |
| int Cfg_Conv(int Filter_order, int En_Conv) | Configure convolution operation length and enable |
| int Cfg_MAC(int Filter_order, int En_MAC) | Configure MAC operation length and enable |
| int Updata_W(int Filter_order, int En_UP.W) | Configure weight coefficient length and enable |
| void Rst() | Reset coprocessor |

The Cfg.Conv library function is taken as an example, and its specific inline assembly syntax format is shown in Figure 10.

```
//   Cfg.Conv
 int Cfg_Conv(int Filter_order, int En_Conv)
{
    int source1=Filter_order;
    int source2=En_Conv;
    asm volatile(
        "Cfg.Conv %[src1],%[src2]"
        :[src1]"r"(source1), [src2]"r"(source2)
    );

  return 1;
}
```

**Figure 10.** Library function of Cfg.Conv instruction.

## 6. Application and Evaluation of the FxLMS Algorithm

### 6.1. Overall Design of ANC System

After completing the hardware and software design of the coprocessor, it is the design part of the whole ANC system. This design is based on Hbird E203_SoC platform, modifies the original SoC, deletes unnecessary peripheral interfaces, and adds IIS interface peripherals needed for audio data transmission. The whole system structure is shown in Figure 11.

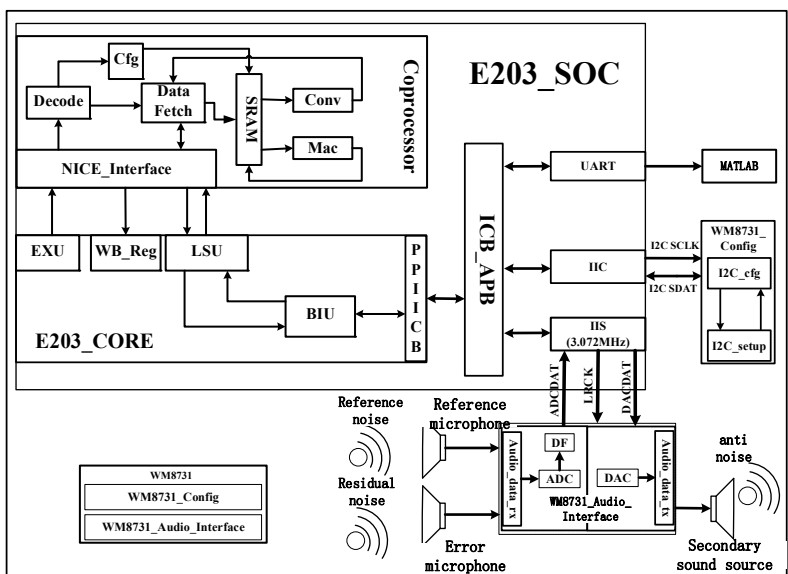

**Figure 11.** Circuit structure diagram of ANC system.

The main processor configures the initial information through the IIC bus to make the WM8731 audio codec module work normally, and uses the probe to collect audio signals and convert them into digital audio signals through ADC built in the module. Then it is transmitted to the ICB bus through IIS audio transmission interface, and the coprocessor reads IIS audio data on the bus through LSU and loads it on the corresponding cache. After configuring enabling instructions, the convolution operation can be carried out smoothly and anti-noise signals can be generated. After that, the anti-noise signal is written to the address where the IIS interface data are located through the memory request channel, and the analog signal is obtained by digital-to-analog conversion through the built-in DAC of the module and secondary noise is generated. Then the module collects the residual noise signal again until it is loaded on the corresponding buffer. After configuring the enabling instruction, the MAC array can update the weight coefficients so as to complete the denoising and acceleration of the ANC system once.

### 6.2. Evaluation Analysis

After the whole software and hardware design and system design are completed, the denoising performance is measured based on MCU200T development board, and the schematic diagram of the measured scene is shown in Figure 12.

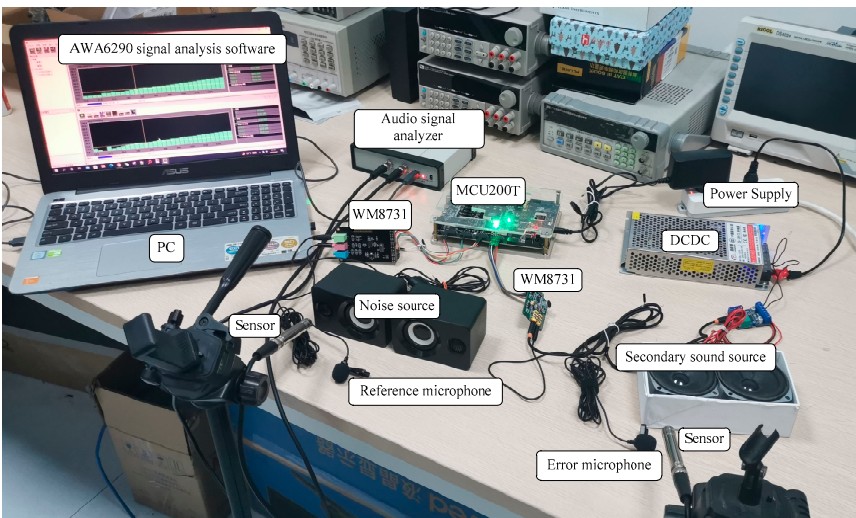

**Figure 12.** Real scene diagram of test scene.

The noise signal is collected before and after denoising by special instruments, and the pink noise is used for acoustic test so as to obtain the relevant collected data and visualize the data through Matlab to obtain the change before and after denoising in the ear frequency band as shown in Figures 13 and 14.

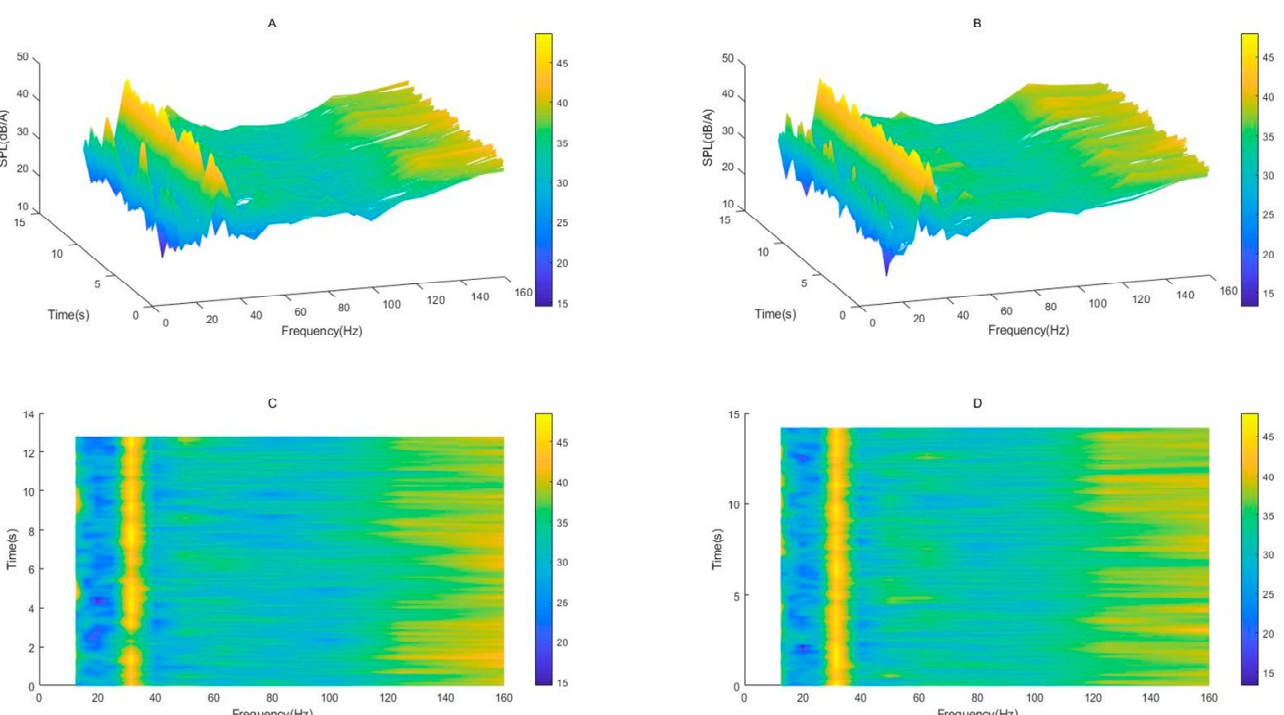

**Figure 13.** (**A**) Change in SPL and time within 0 to 160 Hz before noise reduction, (**B**) change in SPL and time within 0 to 160 Hz after noise reduction, (**C**) time frequency (0–160 Hz) diagram before noise reduction, (**D**) time frequency (0–160 Hz) diagram after noise reduction.

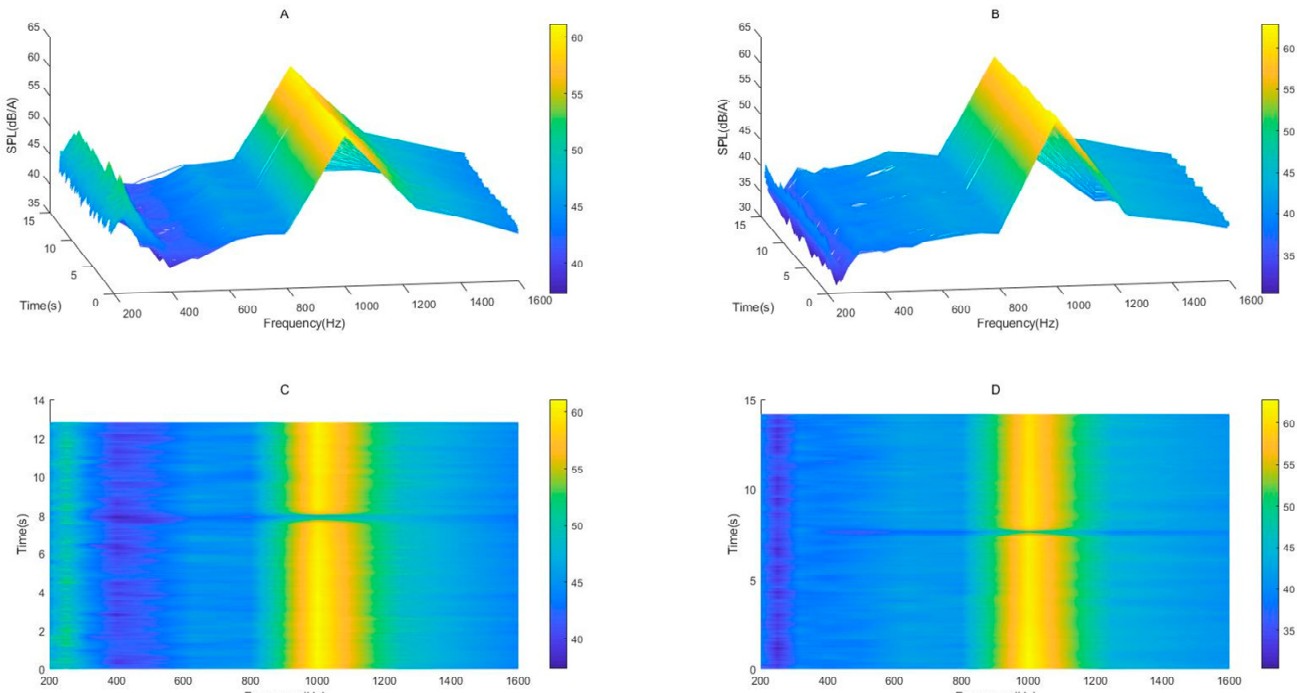

**Figure 14.** (**A**) Change in SPL and time within 200 to 1600 Hz before noise reduction, (**B**) change in SPL and time within 200 to 1600 Hz after noise reduction, (**C**) time frequency (200–1600 Hz) diagram before noise reduction, (**D**) time frequency (200–1600 Hz) diagram after noise reduction.

As can be seen from the Figure 13A,B, the sound pressure levels (SPL) of the same noise are different in continuous time periods, so the denoising effect is unstable. Because their color bar scales are the same, the change in sound pressure level before and after noise reduction is very small, and the change trend is the same. At the same time, it can be seen from the Figure 13C,D that the noise reduction effect is not good at 20 Hz and 60–100 Hz. On the contrary, the SPL after noise reduction is larger (the color of the color bar becomes lighter). There may be two reasons for this result: first, the SPL itself at each frequency point is within a range of change, and the color of the color bar may become darker due to the low sound pressure level at individual points. The second reason is that the FXLMS algorithm itself has a poor noise reduction effect in the ultra-low frequency band.

It can be seen from the Figure 14A,B that the noise reduction effect is obvious at 200 Hz, the curve changes significantly, and the change trend after 400 Hz is similar. However, since the color code of the three-dimensional graph before and after noise reduction is not consistent, the change cannot be seen directly through the color, so it needs to be analyzed in combination with the coordinates of SPL. The noise reduction effect is about 5 dB in the range of 400 to 800 Hz and about 3 dB in the range of 1000 to 1600 Hz. At the same time, it can be seen from the Figure 14C,D that the time-frequency diagram changes significantly at 200 Hz, from light green in Figure 14C to dark blue in Figure 14D; the noise reduction effect at this frequency point can reach 10 dB. Although the color of 400 Hz in Figure 14C is very deep, the sound pressure level at this frequency point is about 42 dB according to its color code. Although the color of 400 Hz in Figure 14D is lighter than that of Figure 14C, its sound pressure level is also about 39 dB, so the noise reduction effect is quite significant. The noise reduction in other frequency points is similar, so it is not be repeated. It should be noted that when the time coordinate axis in the figure is 8 s, the whole frequency band (200–1600 Hz) has a cliff drop, which is caused by the discontinuity of the test noise itself. Therefore, in the description of noise reduction, this part is not included in the calculation of noise reduction effect.

At the same time, it can be known from the Figure 15a that the denoising effect of the algorithm in the middle and high-frequency band is not ideal, which is related to the

denoising principle of the active denoising system itself. According to the relationship between frequency and sound pressure level in Figure 15b and the mean and maximum sound pressure levels of each frequency point in Table 3, noise reduction can reach 8 dB in the frequency range from 200 to 400 Hz, and 5 dB from 500 to 1250 Hz, but decreases to 3 dB from 1250 to 2000 Hz. Because sound pressure level changes incongruently at the same frequency point over a continuous period of time, the optimal noise reduction can reach up to 8 dB in the frequency range from 200 to 2000 Hz. A mean of 5 dB in noise reduction in the frequency range of 200 to 2000 Hz is attainable based on the average sound pressure level, which proves that the algorithm can be realized by combining a heterogeneous SOC with hardware accelerator.

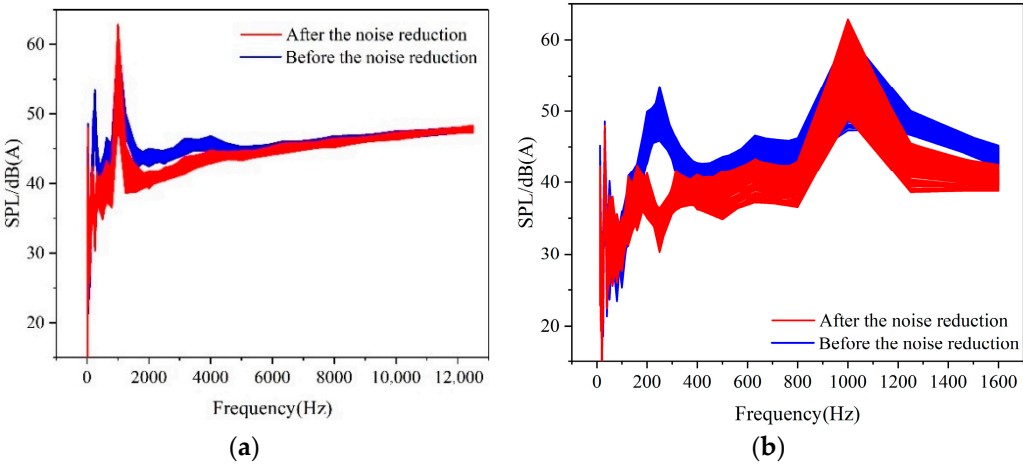

**Figure 15.** Variation in SPL in the frequency. (**a**) Change in SPL within the frequency range of 0 to 12,000 Hz, (**b**) change in SPL within the frequency range of 0 to 1600 Hz.

**Table 3.** SPL at each frequency point before and after noise reduction.

| Hz | 125 | 160 | 200 | 250 | 315 | 400 | 500 | 630 | 800 | 1000 | 1250 | 1600 | 2000 |
|---|---|---|---|---|---|---|---|---|---|---|---|---|---|
| **AVG_F** | 36.5 | 39.5 | 46.1 | 49.7 | 44.1 | 40.4 | 41.7 | 44.6 | 44.4 | 60.4 | 48.5 | 43.8 | 43.7 |
| **AVG_B** | 36.4 | 38 | 39.1 | 33.5 | 39 | 38.7 | 39.5 | 41.6 | 41.1 | 61.3 | 43.9 | 40.7 | 40.6 |
| **Max_F** | 40.9 | 42.7 | 49.9 | 53.5 | 46.3 | 42.8 | 43.6 | 46.6 | 46.2 | 61.1 | 50 | 45.2 | 45 |
| **Min_F** | 31.1 | 34.4 | 42.6 | 45.9 | 41.1 | 37.3 | 38.7 | 41.6 | 42 | 47.3 | 46.7 | 42.1 | 42.4 |
| **Max_B** | 40.8 | 42.4 | 41.3 | 36.3 | 41.6 | 41 | 41.5 | 43.3 | 42.9 | 62.9 | 45.5 | 42.5 | 41.7 |
| **Min_B** | 31.25 | 33.3 | 35 | 30.3 | 36.1 | 36.2 | 34.8 | 37.1 | 36.5 | 48.9 | 38.6 | 38.8 | 39.2 |

In order to evaluate the performance of the coprocessor, this study adopts two methods to implement convolution and MAC operations, one is implemented by the standard RISC-V I\M instruction set, the other is implemented by using the coprocessor custom instructions designed in this study and the RISC-V I\M instruction set together, and compares the number of instructions executed by the two methods. Through IDE tools to write the software code and burn it to the development board, you can print out the corresponding execution results and calculate the number of instructions through the serial port. The experimental results are shown in Table 4.

**Table 4.** Number of instructions required by different arithmetic units to run under different instruction sets.

| Algorithm | Rv32 I\M Instruction | Coprocessor Instruction |
|---|---|---|
| Conv | 4582 | 1324 |
| MAC | 656 | 256 |

Through the instruction number, we can see that Conv and MAC operation can save instruction space more than a standard instruction set under the action of coprocessor, and the instruction number is greatly reduced. This is because on the one hand, the coprocessor realizes convolution and MAC through a special hardware acceleration unit, while the main processor can only realize convolution and MAC through software methods such as addition, subtraction, multiplication, and division; on the other hand, from the system data flow diagram in Figure 8, it can be seen that the coprocessor implementation reduces the repeated movement of data and further improves the processing speed of the algorithm.

## 7. Conclusions

Based on the design optimization of hardware accelerator, The coprocessor was designed, the ANC system was built on the basis of E203_SoC, and the denoising test was carried out in a quiet indoor environment. The sound pressure level data before and after denoising were obtained by audio analysis and acquisition instrument. After data analysis, it can be seen that the FxLMS algorithm realized by combining a heterogeneous SoC with a hardware accelerator has a remarkable effect and can achieve nearly an 8 dB denoising effect. Subsequently, two different test methods were used to test the acceleration effect of the coprocessor, and it was concluded that the implementation of the coprocessor custom instruction set has a significant acceleration effect for convolution and MAC operations.

**Author Contributions:** Conceptualization, J.Y. and W.W.; methodology, J.L.; software, Q.Z.; validation, X.M., Q.L. and Q.Z.; formal analysis, Q.Z.; investigation, X.M.; resources, Q.L.; data curation, J.L.; writing—original draft preparation, Q.Z.; writing—review and editing, J.Y.; visualization, Q.Z.; supervision, W.W.; project administration, W.W.; funding acquisition, J.Y. All authors have read and agreed to the published version of the manuscript.

**Funding:** This research was supported by the Science and Technology Major Project of Chongqing Municipal Science and Technology Bureau (cstc2018jszx-cyztzxX0054), and the Chongqing Municipal Science and Technology Commission Major Project of Integrated Circuit Industry (cstc2018jszx-cyztzx0217).

**Data Availability Statement:** The data presented in this study are available on request from the corresponding author.

**Acknowledgments:** Chongqing University of Posts and Telecommunications are greatly acknowledged for their financial support in making this research possible.

**Conflicts of Interest:** The authors declare no conflict of interest.

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
