# Peer review of "Audio Denoising Coprocessor Based on RISC-V Custom Instruction Set Extension"

_acoustics, doi:10.3390/acoustics4030033_

Round 1

Reviewer 1 Report

This work presents a hardware implementation of the FxLMS algorithm for audio de-noising. The paper focus on the design of a RISC-V based coprocessor and the related software. In order to improve the paper quality some remarks are made.

  • some phrases seem to be too long and difficult to understand (e.g. rows 63-67...)
  • few editing errors and expression must be corrected (e.g.  row 84, 181,104,130 ... fetch fingers? "The main signal is sampled with a reference microphone"...? (no ADC)
  • some citations must to be review e.g. r115 
  • fig.2 may be improved to be more suggestive like in other references
  • fig 3 blue flow trace jump some steps?
  • figs 6,10,12 may be improved to be more readable; fig 11 is no so relevant, may be omitted
  • the steps presented in r278...291 may be included in a flowchart to make it easier to follow
  • all less common acronyms must be specified
  • the experimental setup may be improved
  • r364 "system data flow diagram in Figure 7,...." is wrong Figure 7. is 32-bit custom instruction encoding format.
  • the results must be compared with other similar works or other approaches
  • it is recommended a major review of the paper 

Reviewer 2 Report

  1. Line 108:To be more accurate, the “microphone” should be changed to “error microphone”.
  2. Line 122: Need to be polished, and make it easier for reader to understand.
  3. Line 130-131: Need to be polished, and make it easier for reader to understand.
  4. Line 142-145: I didn’t see gray, pink and light gray. What I can see is gray, pink and light purple. Color need to be reselected in fig. 3.
  5. Line 140: In fig.3, the black flow and blue flow shouldn’t be parallel as shown in fig. 3.
  6. Line 336: Figure 12 needs to be improved as sub-title are overlapped with each other.
  7. In section 6.2, an offline simulation is recommended to compare with test to check if test result can match with simulation result. To be honest, the simulation result should match with test result very well if system works well.
  8. DSP and FPGA are real time system and process data point by point with hardware IRQ, while RISC-V system normally process data with software IRQ. DSP and FPGA have more advantage in ANC data processing than RISC-V, so what is meaning of ANC data processing with RISC-V? The author needs to answer this question clearly and convince reader.

Reviewer 3 Report

The authors designed an audio denoising coprocessor based on the RISC-V custom instruction set extension. The topic of this study is relevant to the field. However, the purpose of this study, methods, and results have not been well presented. Some other serious flaws in this article are as follows: 

  1. When abbreviating a term, the authors should use the full term the first time you use it, followed immediately by the abbreviation in parentheses, e.g. lines 10,11,17,19...
  2. References are missing in many sentences, e.g. lines 54,55
  3. The English wiring is poor and it is hard for me to understand some sentences describing the methods used in this study, e.g. lines 56-59.
  4. The authors described the model and hardware but didn't provide the audience with enough background. The authors should introduce the limitation of previous models and the motivation of the current study, followed by a brief summary of the most important findings in this study.
  5. The most important results of this study was shown in Figure 12. These plots are the results and should not be called "schematic diagrams" in the title. Without quantitative description, it is hard to take the authors' conclusion that "the average denoising effect of 8dB can be achieved in the frequency range of 200-2000Hz".

Round 2

Reviewer 1 Report

The paper was improved and most of requests are solved.

If is possible consider improving the quality of the figures 13,14.

Reviewer 2 Report

thanks for the author's response to the comments and revisions.

Author Response

Thank you for your review and your valuable comments and suggestions.

Reviewer 3 Report

The results are still not clearly presented. The authors should add orders for the subplots in Figure 13, and the results on each plot should be discussed in detail if they were present. Instead, I got lost in the key message that the authors really wanted to show in Figure 13. The front should be larger and clearly seen by the authors.   

Round 3

Reviewer 3 Report

The authors have addressed all my comments and the current version should be good to be published.